# Using Radical Innovation to Overcome Utility Trade-Offs in Urban Rail Systems in Megalopoleis

**Marcelo Blumenfeld \***, **Clive Roberts** and **Felix Schmid**

Birmingham Centre for Railway Research and Education, The University of Birmingham, Edgbaston, Birmingham B15 2TT, UK; c.roberts.20@bham.ac.uk (C.R.); f.schmid@bham.ac.uk (F.S.)
\* Correspondence: m.blumenfeld@bham.ac.uk

**Abstract:** Urban mobility is increasingly becoming accepted as a basic human need, as socio-economic opportunities depend on the ability to reach places within an acceptable time. Conversely, the emergence of megalopoleis as dominant features of the global landscape has increased commuting effort to unprecedented levels, due to the ever expanding urban areas and the associated travel distances. This now poses a risk to the efficient accessibility of cities, but there is an assumption that the problem can be overcome by increasing the speed of transport systems. However, advocates of this approach overlook important utility trade-offs that arise from the conflict between greater vehicle speeds and the additional time required to access the services. In this paper, we investigate this approach and show that higher speeds in metro systems do not always result in faster travel in cities. We then propose a new approach to addressing the problem, which culminates in a solution that can overcome the current paradoxes and increase door-to-door speeds more effectively. The resulting operational concept optimizes speed and coverage in urban rail systems in megalopoleis, accommodating the longer trips within time budgets. We position this research as a starting point to a new perspective on developing complex urban systems in the future.

**Keywords:** urban transport; metro systems; urban rail; travel time budgets; capability engineering

## 1. Introduction

The evolution of societal dynamics around the world is strongly linked to the historical improvements in the ability to move people and goods across geographies. Economic growth and changing social trends, have driven the demand for freight and passenger transport in an almost exponential pattern over the last decades, across all modes [1–3]. These rapid changes in transport volumes witnessed, and the even faster trends forecast for the future, highlight the need for technical and political adaptation at an unprecedented rate [4]. Moreover, Enoch et al. [4] also emphasize the clear disparity between the unchanged operational concepts of transport modes and the flows existing in modern societies. In the current landscape of pressing urbanization and changes in the urban dynamics, it is important to understand these challenges in detail and address the respective issues appropriately. Firstly, it is paramount to understand the influence of passenger traffic on urban transport systems [5]. Secondly, it is crucial to analyze the transport dynamics in the emerging new geographies of urban areas.

The emergence of the megalopolis in the twentieth century, because of increased socio-economic development and technological advances, has resulted in sprawling urban regions where the sustainability of the transport systems becomes an important issue for the new century. As the process of urbanization continues throughout the world, the number and size of urban areas is expected to grow considerably, especially in the developing world, where infrastructure cannot keep up with such accelerate growth. By 2030, forecasts indicate that there will be more than 1.2 billion people living in the more than 100 cities with more than 5 million inhabitants [6]. By the end of the century, it is expected that some cities will house more than 80 million people [7].

The relevance of these urban giants does not lie solely in the magnitude of their populations, but also in their impacts on the global landscape. As complex systems, 'megalopoleis' (plural of megalopolis, Greek, great city) are more than just large cities. They are the economic hubs of the world and emerge because population growth tends to be beneficial to urban areas [8,9]. In short, cities are catalysts of social interaction, accelerating economic productivity and thus creating a positive feedback loop that attracts even more population and stimulates exponential growth. On the positive side, when their population grow, cities tend to benefit from economies of scale in terms of infrastructure requirements, and increased returns in terms of socio-economic indicators [10].

However, the same allometric scaling from the increased interactions also means that externalities, such as pollution, crime, and other negative aspects of urban living, also grow superlinearly. Moreover, their sheer size results in inevitably longer travel distances than exist in their smaller counterparts. In a time when efficient movement within urban areas has become increasingly recognized as crucial to human development and social equity, among other well-established basic human needs, the current spatial arrangements of these areas has instead led to longer travel times and greater environmental footprints [11–13].

The use of the term 'megalopoleis', in this research is intentional, to focus on cities' structures rather than population count. As opposed to the centralized structure of the classical 'metropolis' or 'mother city', a megalopolis is commonly defined as a polycentric urban region comprising various cities and towns that are physically separated, but functionally connected [14,15]. Contrastingly, the term 'megacity' tends to focus only on population numbers, but values can differ by up to 100% between references [16,17]. Finally, even population estimates are unreliable in these cases because of the blurred boundaries between administrative borders and functional units [18].

## 2. Growth of Cities as a Function of Travel Speeds

Megalopoleis could only reach their current size and structure because of advances in transport technologies [19,20]. Since there are limits to tolerable population densities, it is only when transport costs are reduced through higher speeds that cities can expand their boundaries and urban densities decrease [21]. Conversely, when transport links fail, population tends to agglomerate at high densities within smaller areas to maintain access, a feature of eighteenth and nineteenth century cities but still observable in developing countries [21,22]).

The influence of travel speeds on the size of cities is not only well documented in literature, but also has remained remarkably stable over the centuries. Since the amount of time people are willing to dedicate to travel per day (namely, their travel time budget) is limited to around 1.1 h a day, the diameter of urban areas has generally been equivalent to one hour of travel [20,23–27]. Even though some researchers have found utilities in travel times, people would prefer not to invest in longer travel time budgets [28,29].

It logically follows that the faster one can afford to travel within one's time budget, the larger the area of potential exploration available. Under constant time, the radius of an urban areas is then proportional to the speed of travel of the dominant mode [30]. Until the invention of motorized transport, cities were generally limited to a diameter of 5 km, relating to the 'hour-wide' pedestrian access [31]. This changed significantly with the introduction of urban railways. For instance, before the introduction of the railways, London stretched merely a mile from the River Thames, while by the end of the nineteenth century commutes over 5 miles were not uncommon [32,33].

Subsequently, just as the railways transformed cities into metropoleis in the 19th century, the increasing affordability of private motorization and improvements in rail transport have expanded them into megalopoleis in the 20th century. While the maximum speeds allowed on the road network are not necessarily higher than those of segregated rail lines, private modes usually offer higher door-to-door speeds thanks to fewer trip components. Faster operations in commuter rail were essential for denser conurbations, such as London and Tokyo. As a result, they permitted urban areas to stretch much

further and merge into whole urban regions. While a person on foot has a potential area of exploration of 20 km$^2$ within their travel time budget, a person in a car can exceed 1000 km$^2$ within the same time [34]. In fact, these cities have expanded much further than that. Since megalopoleis are polycentric by nature, urban areas can expand beyond the 'one-hour wide' paradigm because trips break from the usual radial pattern. For example, a study found that while the median area of twenty of the world's megalopoleis is 2251 km$^2$, drivers commute approximately 19.7 km each way [22,35]. Figure 1 illustrates the expansion of London, São Paulo, and Tokyo over time.

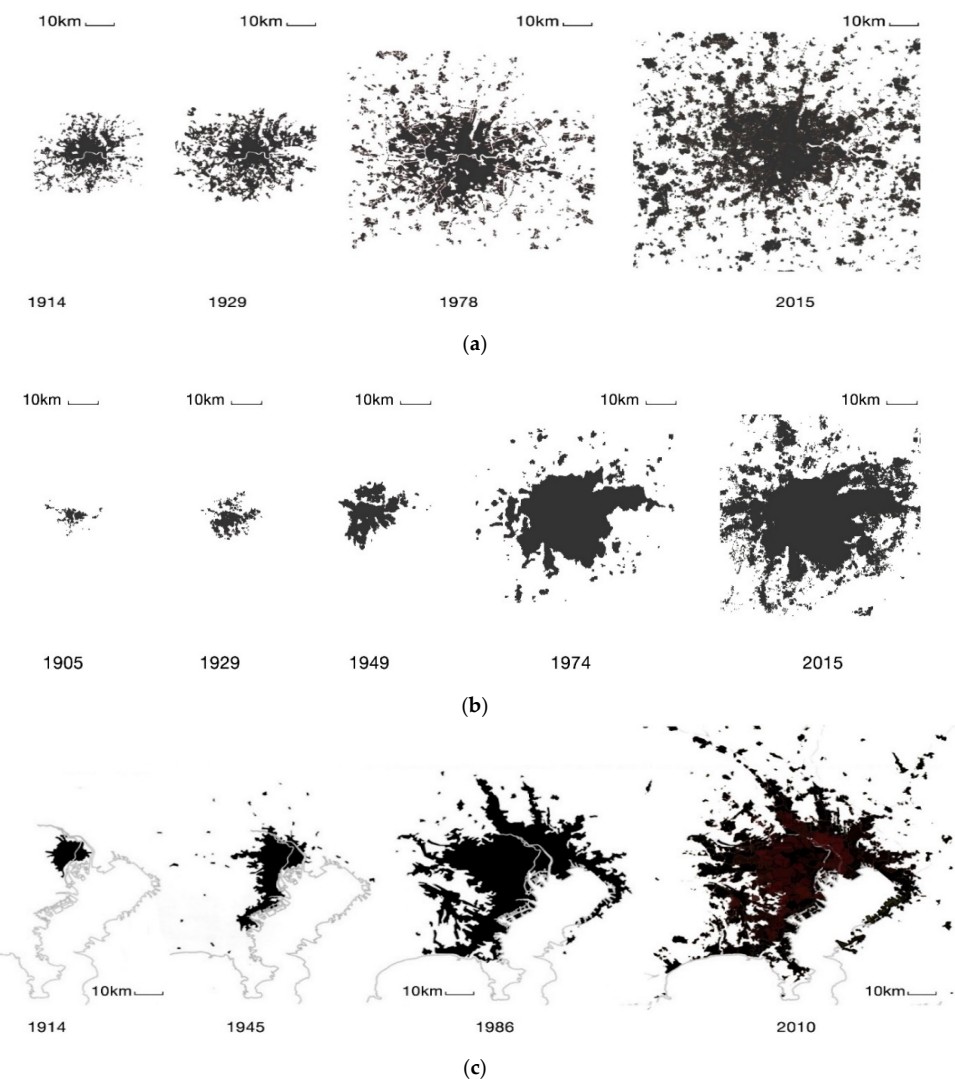

**Figure 1.** Urban expansion of (**a**) London, (**b**) São Paulo, and (**c**) Tokyo over time (authors after ESRI [36] and Okata and Murayama [37]).

One of the consequences of this phenomenon is that travel times in these cities are now exceeding the natural travel time budgets, even giving way to the phenomenon of super-commuting [38]. For instance, Londoners spend 70% longer travelling to work than those in the rest of the country [39]. Similarly, in São Paulo, Brazil's largest urban area, the average person spends around 100 min travelling to work and back, compared to 60 min in smaller cities [40,41]. Tokyo and Osaka, the largest urban areas in Japan, show a similar pattern, where workers spend respectively 97 and 88 minutes in contrast to workers in smaller cities who spend on average 65 min [42].

Moreover, considering that the expansion of many of these urban regions has been driven by the mass adoption of private and semi-public motorized transport, the impact on

travel times is even more severe for those who do not have access to on-demand means of transport [31,43]. In São Paulo, public transport users spend 134 min commuting compared to 62 min spent by those driving [44]. However, the situation is not better in cities with more robust transport infrastructure, as Londoners who travel by rail spend 55% longer than those who drive [39].

It seems clear that megalopoleis have had significant impacts on travel time budgets and the cost of travelling, but the burden is accentuated for public transport users, resulting in either reduced accessibility or longer travel times. It is thus not surprising that some argue that these regions can be reconnected within the normal travel time budgets by increasing the speed of transport modes [26,45]. However, these visions of boundless futures tend to overlook important paradoxes, or apparent contradictions, that arise from the complexity of travel [46].

This new paradigm is relevant to transport research because, for the first time, urban areas have reached a point where urban expansion cannot be simply solved by increasing the maximum speeds of urban rail systems. The extra trip components, added to physical and perceived penalties such as access, interchange, waiting times, and crowding, prevent urban rail from providing a level of service comparable to private modes [47–49]. Whereas overall door-to-door speeds have shown to be crucial to mode choice [30,50], reducing the attractiveness of private modes will not necessarily convert into higher rail use if the perceived travel times and costs of urban rail remain unchanged [51–53].

## 3. Methodology

The research combined the backcasting method with a capability-based approach to work backwards from theoretical end goals towards operational concepts. The backcasting method establishes a theoretical end goal which is transformed into an operational concept using endogenous and exogenous factors. The process then works simultaneously in backwards and forwards directions, modelling the current capabilities of the system against the selected goals with the aim of developing an operational concept using a capability approach.

Figure 2 illustrates the method adopted, using a data flow diagram (DFD). The main logic behind the process is the normative (value judgement based) perspective of backcasting towards the future. As explained by Robinson [54], it works backwards from a desired end state and identifies the necessary steps to achieve such a goal. Transferred to a technical process using a capability-based approach, the process consists of four main steps that help to understand a given problem and to identify potential solutions.

In the context of the utility trade-offs of urban rail systems, the goals are defined by the parameters of door-to-door journey times and accessibility. These exogenous factors have been extensively covered in research to provide measures of effectiveness of systems in achieving them. Similarly, the assessment of current capabilities uses both validated models and a review of previous solution attempts to identify the critical points of inherent trade-offs and paradoxes.

From there, the future looking method enables a flexible approach to performance, opting for capability-based on the logical technological advancements expected over a certain timespan. This means that the operational concept developed prioritizes a new solution to the trade-offs over current technical capability. This is where the normative approach to problem solving overcomes existing trade-offs with a capability driven operational concept.

Once an operational concept has been developed, it then becomes possible to reverse engineer designs for the solution using modelling techniques. The process, developed by the International Council of Systems Engineering (INCOSE) [55], uses modelling to create a level of 'anchoring' in solutions that are realistic and likely to be feasible within the given timeframe. This also enables practitioners to derive more specific requirements and performance parameters for the design, which in turn feed an iterative review of the operational concept.

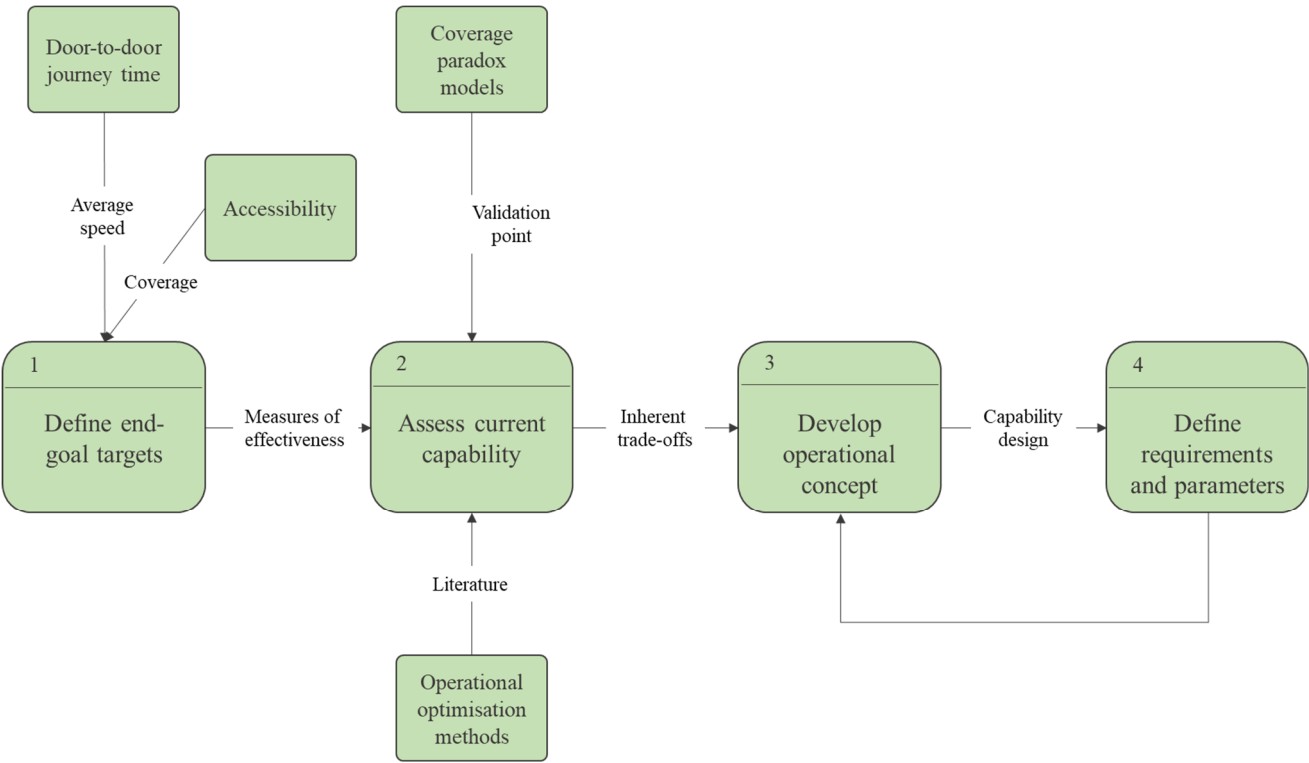

**Figure 2.** Data flow diagram (DFD) of the methodology.

## 4. Inherent Trade-Offs in Urban Rail Trips

Door-to-door travel times are inherently limited by trade-offs in all motorized modes, so moving more quickly does not necessarily convert into shorter travel times. In the case of private modes, such as cars or motorcycles, maximum speeds and consequently travel time are limited by traffic density and speed restrictions, amongst other elements [56]. That way, in a megalopolis where private modes are the main form of travel, these are not only harmful to the environment but also cannot accommodate the large demand in networks with such limited capacity. In public modes, such as buses or trains, door-to-door travel time depends on both the time to access the mode and the in-vehicle travel time, among other elements.

These trade-offs have profound implications for the perception of urban boundaries. Regardless of the maximum speed achieved at a certain point, it is the average door-to-door speed, and thus the overall journey time, that defines the city limits for a particular user of a transport mode within their travel time budget. For these and other economic reasons, airplanes have never been able to create commutable regions of 500 km radius because of the time spent on the non-flying components of the trip. Similarly, although the high-speed rail route between Tokyo and Osaka is practically entirely urbanized, commuting efforts in terms of door-to-door time still prevents the corridor from forming a single urban region.

As evident from Equation (1) door-to-door generalized travel 'cost', expressed as a time, consists basically of five parts: (i) Access time ($T_a$); (ii) entry and exit times ($T_e$); (iii) waiting time ($T_w$); in-vehicle time ($T_v$); and (v) interchange time ($T_i$). For our calculations, we assume the access time $T_a$ to be the average time spent covering the distance to and from the means of transport at both ends of the journey, and $T_e$ to be the average time between ticket barriers and platforms, and vice-versa. Specific weightings are added for the relative value of time ($\gamma$, $\delta$, $\varphi$, $\omega$) compared to in-vehicle time.

$$T_t = \gamma 2T_a + \delta T_w + T_v + \varphi 2T_e + \omega T_i \tag{1}$$

A driver, for example, would have a relatively short access time ($T_a$) from door to car and from car to door, no waiting or entry/exit times ($T_w$ or $T_e$) as the vehicle is readily available, and the trip mainly consists of an in-vehicle component ($T_v$). It can also include interchange time ($T_i$) if the trip will also comprise another motorized component. Conversely, the trip on an urban rail system is more complicated: Access time ($T_a$) from door to station and from station to door; entry and exit times ($T_e$) from the station door to the platform; waiting time ($T_w$) for the time at the platform; in-vehicle time ($T_v$), which comprises the time spent inside the vehicle; and interchange time ($T_i$), which accounts for the time to change between lines and/or between modes.

More importantly, trip components interact with each other often in conflicting ways. From the literature, we know that $T_v$, illustrated in Equation (2)) is the sum of all times spent travelling between each station for $n$ stops, at maximum line speed ($V$, in m/s), during acceleration ($\alpha$, in m/s$^2$), braking ($\beta$, in m/s$^2$), and dwell time ($T_d$). Jerk time ($T_j$) accounts for the time needed to comfortably transition to acceleration and braking. Consequently, it depends considerably on the distance between stations (D) [57]:

$$T_v = \sum_{i}^{n-1} \sum_{j}^{n} \frac{D_{ij}}{V} + \frac{V}{2}\left(\frac{1}{\alpha} + \frac{1}{\beta}\right) + nT_j + T_d \tag{2}$$

From these equations, our models show that there are inherent trade-offs that arise when trying to increase in-vehicle speeds and thus limiting the efficiency of urban rail systems in the context of megalopoleis. For our calculations, we will adopt a journey distance of 19.7 km, and normal operational parameters of metro lines as listed in Table 1 [58–60]. We assume that users walk to stations and have an average of one interchange per journey and user.

**Table 1.** Operating parameters of metro systems.

| Parameter | Value |
|---|---|
| Maximum line speed ($V$) | 25 m/s (90 km/h) |
| Acceleration rate ($\alpha$) | 1.3 m/s$^2$ |
| Braking rate ($\beta$) | 1.2 m/s$^2$ |
| Jerk rate ($j$) | 0.75 m/s$^3$ |
| Interchange time between lines ($T_i$) | 270 s |
| Waiting time ($T_w$) | 60 s |
| Entry/exit time ($T_e$) | 165 s |
| Dwell time ($T_d$) | 30 s |

Many scholars have thus investigated the subject with the aim of minimizing the disutilities of trade-offs and reduce the generalized travel costs of public transport systems that pose barriers to their use [48,49,57,61]. Nonetheless, as we will discuss below, these come accompanied by inherent trade-offs in either travel times or wider impacts on urban structure and social, economic, and environmental sustainability.

### 4.1. Reducing the Number of Stops

Without changing any operational parameters, the only way to increase in-vehicle speeds is to change the distance between stations. One way to achieve that is by reducing the number of stops along the line. Givoni and Rietveld [49] explore this scenario for commuter rail trips into Amsterdam, based on findings that users were not necessarily always using their closest station. However, this in turn increases the access distance (d) that users must travel to use the service and, since access time has a higher value of time than in-vehicle time [62], the increase in generalized costs are expected to be substantial. We assume the access distance (d) to be half of the distance between stations (D), as shown in Figure 3 (which was generated using random computer models for illustration). It is

recognized that there is usually an overlap between catchment areas, but this is generally counterbalanced by those travelling further than *d* to access the station.

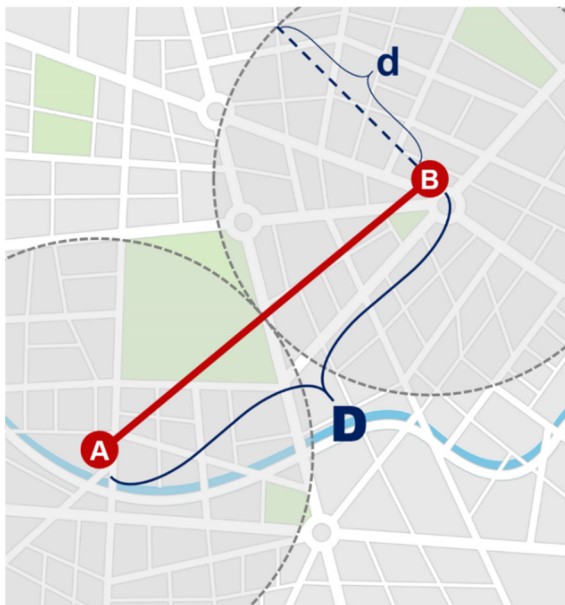

**Figure 3.** The relationship between interstation distance (**D**) and access distance (**d**). (Authors).

Figure 4 illustrates how these trade-offs prevent changes in interstation distances from reducing door-to-door travel times. In this case, the minimum travel time achievable by urban rail is approximately 47 min, 42% longer than the time spend by drivers found by Gyimesi et al. [35].

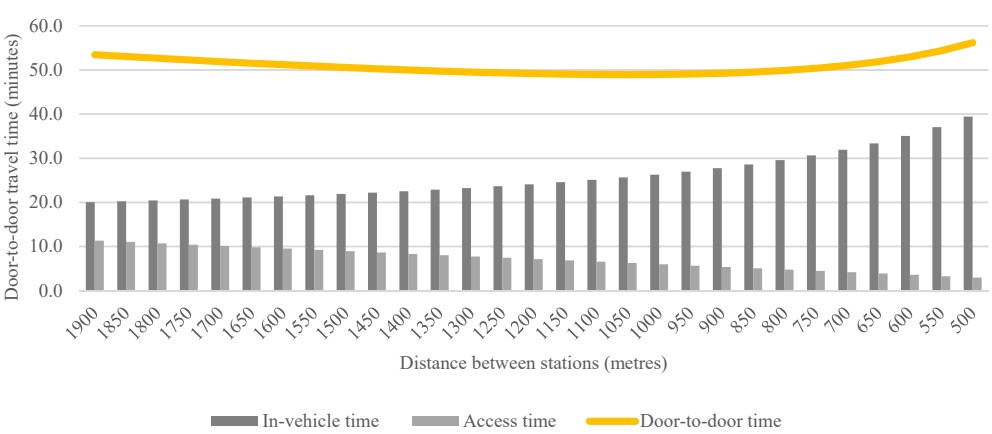

**Figure 4.** The trade-off between access time and interstation distances on a 19.7 km journey. (Authors).

Consequently, reducing the number of stops does not necessarily result in shorter travel times because of the conflict between access times and line speeds, and important considerations arise from it. Firstly, each 100 m increase in interstation distance (D) reduces interstation time ($T_s$) by 4 s but adds an extra 40 s to access time ($T_a$). Considering that the value of time for walking is usually 1.7 times that for in-vehicle time, it means that fewer stops are in fact more counterproductive than reducing in-vehicle speeds [49,62]. Secondly, the value of access time also increases when distances become longer, mostly because these trips will likely include interchanges and certain aspects of unreliability [62]. Finally, greater interstation distances may require access by motorized modes, imposing extra interchange times and barriers to those who do not or cannot drive [48].

### 4.2. Increasing Maximum Line Speed

From an access and social equity perspective, it is better that stations are as close to each other as possible. It is well known that property prices around stations are higher, so limited stations along the line can impose higher disutilities to those who are more likely to depend on public transport [63–65]. Moreover, research shows that adding stations to a line can induce densification, public transport use, and economic development around the extra stations [66–68]. Taking the general elements of urban dynamics into account, urban rail transport systems should focus on maintaining short interstation distances to promote a more balanced and sustainable polycentric distribution. This enables all able bodied users to access stations by walking or cycling, thus reducing social, economic and environmental issues and increasing the health of individuals. In addition, shorter distances between stations promote denser regional development, which improves accessibility with lower mobility needs [69]. Where more detail is required, one can model the minimum interstation distances by adding the distances required for acceleration, braking, and jerk. For calculation purposes, we have overlooked track equipment such as switches and crossings, and line-side equipment that might require speed limits.

Yet, the results provided in Figure 5, again for the 19.7 km journey, show that increasing maximum line speeds with minimum interstation distances is also unlikely to solve the trade-offs and provide door-to-door travel times within the normal travel time budgets, given the distances to be covered and the number of stops on the line. Since acceleration and braking are limited for passenger comfort, increasing line speeds will inevitably require longer access time, due to longer distances to accelerate and brake. With that, the same problems arise as above, and the minimum door-to-door travel time achievable is approximately 45 min even when maximum line speed is 135 km/h. In that case, access time would be approximately 8 min, but the number of stops prevents the system from achieving a more efficient door-to-door travel time. Moreover, on top of these issues, higher operating speeds also raise concerns over the increased energy consumption of such a system.

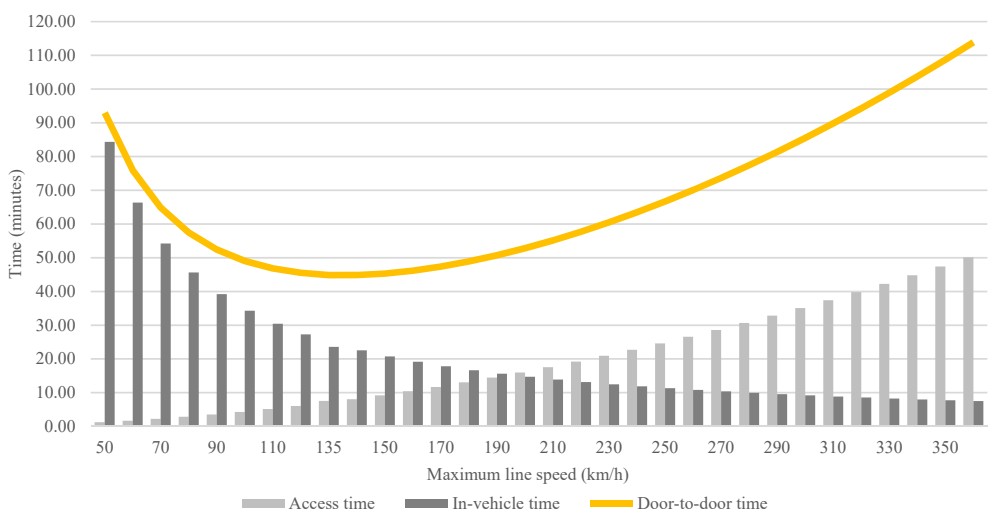

**Figure 5.** Door-to-door travel times based on maximum line speeds for a 19.7 km trip. (Authors).

For instance, a journey by train involves a trade-off between two components: individual access ($T_a$) to and from stations (because walking and cycling speeds are limited by physical ability), where distance to the station is the critical factor; and in-vehicle time ($T_v$), where average line speed is the critical factor. Hence, in order to increase in-vehicle speeds, stations need to be far apart. However, for a more accessible public transport system, where access times must be short, stations need to be closer together.

### 4.3. Accelerated Operations

In face of these systemic shortcomings of mass transport systems, several studies have proposed strategies and technologies to increase average speeds without increasing the distance between stations. In terms of accelerated operation strategies, Furth and Day [61], Fu et al. [70], and Vuchic [57] suggest three methods to increase speeds at constant interstation distances: (1) local/express; (2) zonal; and (3) skip-stop. These methods all share the same principle of serving only some of the stations on the line and thus limiting the number of undesired stops for users and reducing in-vehicle travel times.

Local/express services, perhaps the most common, comprise two different service types: one that stops at every station (local service), and one that skips some of the stations and stops at those with higher demand (express service) [57]. Lines can have different types of express services in order to cater for different demands, such as the Nankai line in Osaka that offers five different stopping patterns along the same line. Zonal operations, as the name suggests, divide the urban area in distinct zones and trains connect each zone to the center. Journey time, consequently, is reduced as the train omits stops at stations in the zones between the center and the destination zone. Finally, skip-stop operations assign stations of a line into three categories: A, B, and AB; while trains are run in two different patterns: Trains A stop at A and AB stations and trains B stop at B and AB stations.

Although all strategies can reduce the fleet size and operational costs, they have significant influences on the generalized cost of users. Firstly, they will create significant disparities in travel times between users in different locations. In addition, the appraisal of demand based on value of time can reinforce these disparities providing event better access to those who already enjoy good access. For instance, Lee [71] found that, after the adoption of skip-stop operations on line 4 of the Seoul Metro, while in-vehicle times became 20% to 26% shorter, waiting, transfer and additional access time increased by 24% to 38%. Secondly, since overall frequency tends to be reduced, the service tends to operate below its capacity, which is an important factor for metro systems in large cities. Finally, such patterns predict savings on linear journeys where interchanges are limited, such as the case of centralized urban areas. Furthermore, the polycentric nature of megalopoleis and the resulting complexity in travel patterns means that accelerating strategies reduce the points of interchange for users and thus add important penalties to travel time and accessibility.

### 5. A Novel Approach

From a systems perspective, it becomes evident that the paradigm created by the emerging structures of megalopoleis requires a new approach to the problem. Moreover, under changing conditions, such as the rapid urbanization in the developing world, adaptation can only be achieved by a radical redesign leading to a completely new internal structure [72]. It requires us to consider the system as a whole, if we are to improve door-to-door efficiently in these large urban areas. On the other hand, the current stage of infrastructure provision in developing megalopoleis permits more innovative solutions with a future-oriented approach so as to leapfrog the gaps between present supply and future demand.

Therefore, we propose here a novel strategy that benefits from being conceptual, as it enables us to find a solution that overcomes the coverage paradox without the trade-offs of the current systems. From a holistic perspective, modelling an operational concept is a helpful step in innovation, as it envisions a totally bespoke system that is fit-for-purpose instead of the ineffective incremental process adopted so far.

Firstly, each service stops at stations along the line observing a certain pattern ($P_x$). If a line is either circular or operated as if it were an infinite loop and when the number of stations is not divisible by the number of patterns, our mathematical model shows that each vehicle will eventually stop at every station, taking a number of 'laps' equal to its pattern in order to call at all stations. A service that stops every three stations ($P_3$) will take three laps to serve all stations on the line, regardless of the total number of stations.

The proposed concept solves the issues of local/express operations as all stations are equally served by all patterns, thereby providing equally optimal travel times to all users, since any origin-destination journey on the same line requires at most one change. The extra time from the additional change is only needed for prime number intervals between origin and destination. We adopt this as platform interchange time ($T_p$) and assume it to be equal to the headway between trains.

Secondly, when stations are located off the main line, the distance between them can be reduced to a minimum without affecting operational speeds. Vehicles that are not serving a particular stop will continue on the main line while those stopping will move to the loop and start braking (Figure 6). When reaching a platform at any station, passengers can change to the vehicle or vehicles that serve their destination station.

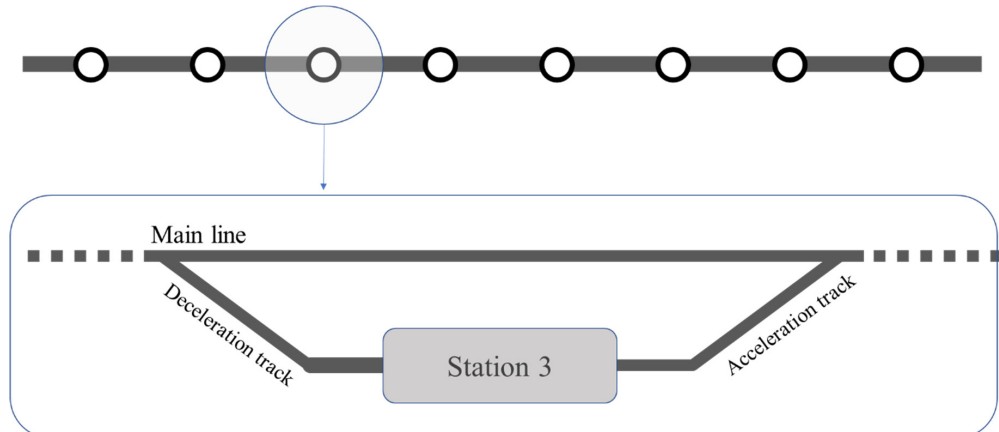

**Figure 6.** Illustration of off-line stations. (Authors).

Consequently, the closer station spacing maintains the access time at its minimum while the optimized operational model increases the average in-vehicle speeds. For each journey distance, there is an optimal combination of maximum line speed and access time that results in the minimum door-to-door journey time. Figure 7 illustrates the average door-to-door times for the same 19.7 km journey using five different patterns: stopping at every station, and stopping at every second, third, fifth and seventh stations ($P_{1,2,3,5,7}$) respectively.

Our model shows that, thanks to the different service patterns, passengers can cover the 19.7 km in approximately 34 min when the maximum line speed is 80 km/h, similar to the speeds of metro systems currently in operation. In this arrangement, stations are 735 m apart, on average, and access time is under 5 min. Therefore, this solution provides a better performance than higher speed lines with high access times and enables users to travel even longer distances within their natural travel time budgets with better accessibility.

This holistic approach can also create a normative perspective that influences the design of subsystems and components that increase the efficiency of transport systems based on specific requirements obtained in the model. For example, entry/exit times ($T_e$) and interchange times ($T_i$) were reduced to two minutes through improvements to station design, thus the door-to-door journey time could be reduced to 30 min, aligned to the 'one-hour wide' constant of cities. Furthermore, such an arrangement produces a flexible model, where operational requirements are dependent on distances and patterns collected from users. With that, there is a degree of scalability to the strategy in the face of specific cases in different cities.

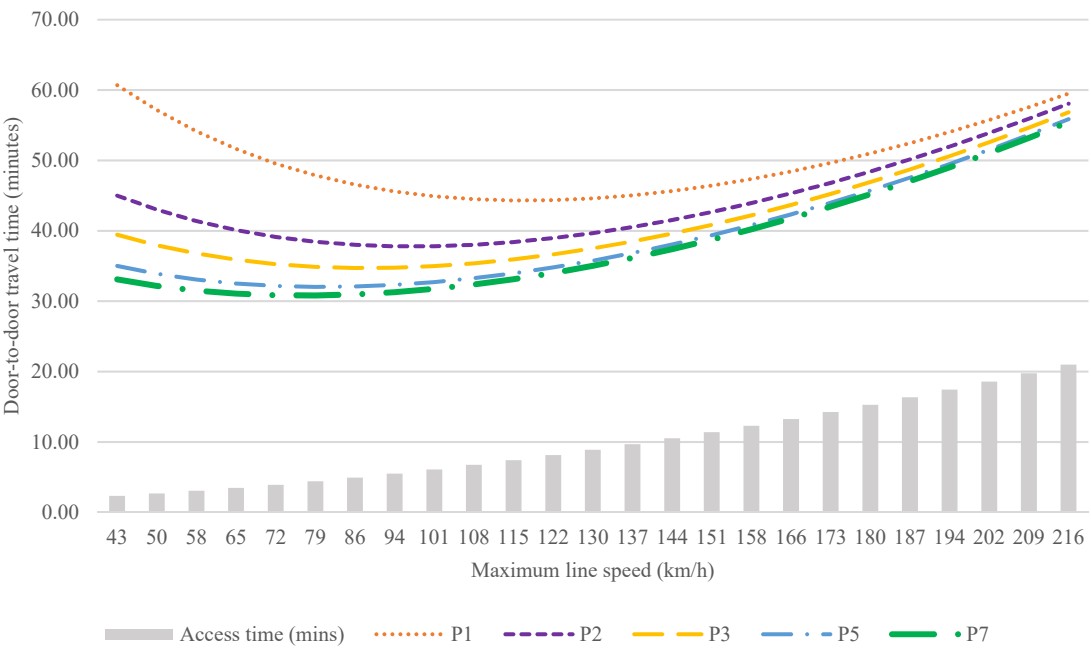

**Figure 7.** Door-to-door journey times for the new strategy under normal metro parameters. (Authors).

## 6. Discussion

The main argument of this paper is that solutions need to apply a systemic perspective to the problem of inherent utility trade-offs of current operational models in urban railways, and examine all components simultaneously, with the added benefit of scalability for different travel distances. We propose an operational concept that can potentially overcome the coverage paradox with technological requirements that are similar to those of current metro systems, thus promoting a potential solution to urban areas undergoing rapid expansion.

The trade-offs between coverage and door-to-door speeds have been a prominent challenge for urban transport operations. Givoni and Banister [46] and Blainey et al. [48], mentioned in the analysis, emphasize the limitations of converging maximum line speeds into door-to-door speeds. Just as the challenge has been pervasive over the decades, attempts at solving such a paradox date back to the 1970s, when Vuchic [73] proposed skip-stop operations to overcome the utility trade-off.

The solution proposed here utilizes aspects of the same principle of optimizing door-to-door journeys that advances the efficacy with a solution that incorporates an outcome based target, rather than projecting greater performance gains. The result solves the different accessibility levels that skip-stop operations create (where some users may need to travel backwards to reach an appropriate service), and also allows for shortest interstation spacing based on user journey times and not on operational performance.

Expectedly, questions may arise regarding the cost and space requirements of such arrangements. In terms of infrastructure requirements, this operational concept lies between a regular double-track system and a four-track arrangement of local/express operations. A four-track layout is needed around the stations for the operation of different speeds, while the areas in between can be reduced to double-track where all vehicles travel at the same speed. Retrofitting existing infrastructure may prove complicated, how-ever, this operational concept may offer potential solutions for cities where railway infra-structure already exists.

Similarly, the operational concept hereby proposed demonstrated stability in simulated environments, yet this will need validation in other laboratory setting and further technology readiness level (TRL) domains. While rapid progress is being seen on closer running in railway systems, headways achieved in this solution have yet to be tested. Even

with close crossings needed at stations, simulations have shown stability under certain uncertainties in the human–machine interfaces.

All in all, the operational concept presented benefits from higher flexibility in its application to the specific context of each city. As operational requirements are dependent on travel patterns, the system can be adapted to users' behavior rather than attempting to force the users to adapt to the characteristics. In addition, the freedom from technical parameters at this stage means that the model can be applied to different modes and not only urban rail. For instance, the reader may see how this strategy would benefit bus rapid transit systems or personal rapid transit systems.

Further information on backcasting can be found in Robinson [74] and on systems thinking in Blanchard and Fabrycky [75].

### 7. Conclusions and Further Work

The starting point of this paper was the emerging transformation of the urban landscape and the recent debate on how to reinstate the equilibrium in travel times in megalopoleis. Their expansion as a result of the reduced cost of travel permitted by technological advancements now challenges the sustainability of transport systems to provide access within reasonable travel times. As travel distances in these urban giants continue to grow, the need for faster travel becomes a key point for the century when the size and number of megalopoleis is only expected to increase.

However, travel speeds of motorized modes of transport are inherently limited by trade-offs that arise from conflicting components, which appear to be highlighted by the emergence of megalopoleis. Strategies to increase travel speeds picked from the literature seem to have limited impact as they also create further trade-offs in the already sensitive values of time of access, waiting, and interchange.

The proposed novel strategy is based on services that stop in different patterns, namely every station, and every two, three, five, and seven stations, which can reduce in-vehicle times without adding extra penalties to other trip components such as access time. The solution differs from skip-stop and local/express operations as all services attend all stations, ensuring a better distributed access throughout the line.

The operational concept presented demonstrates a possible solution to overcome the utility trade-offs that inhabit the paradoxes between speed and coverage in urban rail systems. It is seen as a starting point for further work in enabling the technical requirements in higher TRL domains. For instance, the need for vehicles to change routing in a much faster time window than the current capabilities of S&C systems can initiate endeavors on radical innovation and/or optimization. Similarly, there is an opportunity for the application of signaling and control systems currently under development in other fields.

In conclusion, this research sheds light on the need for a different approach to problems that require radical innovation, either due to internal systemic conflicts or changing environments. When current trends lead to undesirable paths, a normative stance to engineering is needed to direct solutions to achieve a desired point in the future. Operational models can guide the technological development of components and subsystems so that they satisfy the whole system requirement rather than create new trade-offs. On that matter, the conceptual nature of the proposed strategy is intentional so that it becomes possible to draw the necessary technical and technological requirements to overcome systemic barriers. With that, we anticipate a starting point for further research into transport technologies that uses such systemic approaches to develop solutions that can efficiently fulfil the future needs of urban regions.

**Author Contributions:** Conceptualization, M.B., C.R. and F.S.; methodology, M.B., C.R. and F.S.; validation, M.B., C.R. and F.S.; formal analysis, M.B.; investigation, M.B.; writing (original draft preparation), M.B.; writing (review and editing), M.B., C.R. and F.S.; visualization, M.B.; supervision, C.R. and F.S.; project administration, M.B.; funding acquisition, M.B. All authors have read the published version of the manuscript and agreed to its publication.

**Funding:** This research was funded by CAPES, grant number 99999.013598/2013-09.

**Institutional Review Board Statement:** Not applicable since the study did not involve humans or animals.

**Informed Consent Statement:** Not applicable.

**Conflicts of Interest:** The authors declare no conflict of interest.

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
