# Peer review of "Using Radical Innovation to Overcome Utility Trade-Offs in Urban Rail Systems in Megalopoleis"

_futuretransp, doi:10.3390/futuretransp1020010_

Round 1

Reviewer 1 Report

Page 3, Figure 1: Please indicate in your Figure which map corresponds to London, São Paulo, and Tokyo.

Please provide the source for the geographical background of the map in Figure 2 (page 6).

Figure 6 (page 9) must be Figure 5 and Figure 5 (page 10) must be Figure 6.

Page 10, Section 5. Discussion and further work: My suggestion is to include a paragraph where you can discuss whether your findings comply with the findings of similar works found in the literature or not (and if yes, to what extent). In addition, you can provide the necessary references. In the same Section please summarize the limitations and constraints of your research. Finally, please try to address your policy recommendations to the stakeholders involved.

My suggestion is to include a Data Flow Diagram (DFD) which will include all the methodological steps of your research. This DFD will help the reader to obtain a clear overview of your work from the early beginning of your paper. Perhaps this DFD can be placed in Section 3. Inherent trade-offs in urban rail trips.

It seems that References [66] and [67] are not included in the main body of the manuscript although they both appear in the List of References.

Author Response

Dear reviewer,

Thank you very much for your fruitful suggestions to our paper. We have revised the manuscript, and now believe that it encompasses the points made, as below:

  1. Page 3, Figure 1: maps identified with their respective cities
  2. Geographical background of Figure 2 (which is now Figure 3) explained on line 232
  3. Figure numbers corrected
  4. Sections divided into two parts: Discussion, and Conclusion and Further Work. The Discussion covers the relationship with previous works in the paragraphs between lines 387 and 398.
  5. A Data Flow Diagram (Figure 2) was added to a new Section 3 (Methodology)
  6. References were reviewed and corrected

Reviewer 2 Report

This study focuses on utility trade-offs in urban rail systems in megalopoleis. I think the paper fits well the scope of the journal and addresses an important subject. However, a number of revisions are required before the paper can be considered for publication. There are some weak points that have to be strengthened. Below please find more specific comments:

*The abstract could be expanded a bit. In particular, I suggest adding a sentence or two highlighting contributions and outcomes of this work.

*The authors start the introduction section with a brief discussion regarding the emergence of the megalopolis and urbanization. This discussion should be improved. In particular, the authors should start the introduction section with a more generic discussion regarding the growing volumes of passenger and freight transport around the globe and their important role for economic development of different countries and major cities. This discussion should be supported by the recent and relevant references, including the following:

  • Belokurov, V., Spodarev, R. and Belokurov, S., 2020. Determining passenger traffic as important factor in urban public transport system. Transportation Research Procedia, 50, pp.52-58.
  • Dulebenets, M.A., 2020. An Adaptive Island Evolutionary Algorithm for the berth scheduling problem. Memetic Computing, 12(1), pp.51-72.
  • Enoch, M.P., Cross, R., Potter, N., Davidson, C., Taylor, S., Brown, R., Huang, H., Parsons, J., Tucker, S., Wynne, E. and Grieg, D., 2020. Future local passenger transport system scenarios and implications for policy and practice. Transport policy, 90, pp.52-67.
  • Pasha, J., Dulebenets, M.A., Kavoosi, M., Abioye, O.F., Theophilus, O., Wang, H., Kampmann, R. and Guo, W., 2020. Holistic tactical-level planning in liner shipping: an exact optimization approach. Journal of Shipping and Trade, 5, pp.1-35.
  • Albayrak, M.B.K., Özcan, İ.Ç., Can, R. and Dobruszkes, F., 2020. The determinants of air passenger traffic at Turkish airports. Journal of Air Transport Management, 86, p.101818.

After this discussion, it would be logical to discuss the emergence of the megalopolis and urbanization specifically. This will significantly improve the flow of section 1.

*The authors discuss most of the relevant literature in section 2. I suggest adding a paragraph that summarizes the state-of-the-art, identifies the existing gaps, and how these gaps are addressed by the present study.

*Page 4: Please provide a reference for equation (1) and other equations throughout the manuscript. This will help justifying the use of these equations.

*The presentation of the manuscript could be improved by creating two separate sections: one devoted to “discussion”, while the other devoted to “conclusions and future research”.

Author Response

Dear reviewer,

Thank you for taking the time to read our paper and provide fruitful comments. We have revised our manuscript and believe to have addressed the recommendations, as follows:

  1. Abstract has been expanded
  2. An introductory paragraph was added to the Introduction section, addressing the growing transport volumes and economic development.
  3. Discussion section divided into two sections: Discussion, and Conclusion and Further Work
  4. A commentary on the research in relation to previous work and the gaps in the state-of-the-art has been added to the Discussion section
  5. Equations have been referenced and justified throughout the text.

Round 2

Reviewer 2 Report

The authors took seriously my previous comments and made the required revisions in the manuscript. The quality and presentation of the manuscript have been improved. Therefore, I recommend acceptance.